# Chemical Composition of Aerial Parts Essential Oils from Six Endemic Malagasy *Helichrysum* Species

**DOI:** 10.3390/plants9020265

**Published:** 2020-02-18

**Authors:** Delphin J. R. Rabehaja, Guillaume Bezert, Stéphan R. Rakotonandrasana, Panja A. R. Ramanoelina, Charles Andrianjara, Ange Bighelli, Félix Tomi, Mathieu Paoli

**Affiliations:** 1Département de Phytochimie et Contrôle Qualité, Institut Malgache de Recherches Appliquées (IMRA), B.P. 3833, 101 Antananarivo, Madagascar; delphin.rabehaja@imra.mg (D.J.R.R.); charles.andrianjara@imra.mg (C.A.); 2Laboratoire Sciences Pour l’Environnement, Université de Corse-CNRS, UMR 6134 SPE, Route des Sanguinaires, 20000 Ajaccio, France; guillaume.bezert@sfr.fr (G.B.); bighelli_a@univ-corse.fr (A.B.); paoli_m@univ-corse.fr (M.P.); 3Department of Ethnobotany and Botany, National Center for Applied Pharmaceutical Research, 101 Antananarivo, Madagascar; dep.bot.cnarp@mesupres.edu.mg; 4Laboratoire des Industries Agricoles et Alimentaires, Ecole Supérieure des Sciences Agronomiques, Université d’Antananarivo, B.P. 175, 101 Antananarivo, Madagascar; panjarama@moov.mg

**Keywords:** endemic *Helichrysum*, Madagascar, essential oil

## Abstract

The essential oils of six endemic Malagasy *Helichrysum* species were investigated by GC (RI), GC–MS and ^13^C NMR spectrometry. In total, 153 compounds were identified accounting for 90.8% to 99.9% of the total composition. The main constituents were α-pinene for *H. benthamii,* 1,8-cineole for *H. dubardii*, (E)-β-caryophyllene for *H. indutum,* and *H. bojerianum*. *H. diotoides* essential oil was characterized by the presence of two lilac alcohols and four lilac acetates whereas *H. hirtum* essential oil exhibited an atypical composition with 7β-*H*-silphiperfol-5-ene, 7-epi-subergorgiol, and 7-epi-silphiperfol-5-en-13-oic acid as major components.

## 1. Introduction

Madagascar, a big island in the Indian Ocean, is one of the countries in the world having particular hotspot biodiversity. Together with this biological richness, medicinal plants hold an important place in the everyday life of Malagasy people. The medicinal plants inventoried in Madagascar consist of 3245 species, of which 60% are endemic. *Croton* L. and *Helichrysum* Mill. are the most represented genera.

*Helichrysum* Mill. [1] is a large genus of the Asteraceae family and 112 species are known in Madagascar among which 46 are endemic [2]. Essential oils and extracts are obtained from the whole plant or from different parts of the plant and they are used in perfumery and aromatherapy. Biological activity or insecticidal activity have been reported [3]. In the literature, the chemical composition of essential oil (EO) from seven species has been studied [3,4,5,6,7,8,9,10,11]. *H. faradifani* and *H. gymnocephalum* are the most studied species and four papers reported their chemical compositions. Three different compositions are described for *H. faradifani* essential oil: (i) (E)-β-caryophyllene (34.6%) [6], (ii) β-himachalene (15–32.8%) associated with α-fenchene (13.1–27.3%) [5], and (iii) α-fenchene (32.3% and 35.5%) [3,10]. The chemical composition of *H. gymnocephalum* oil is homogeneous and characterized by the occurrence of 1,8-cineole (66.7% [7], 59.7% [6], 47.4% [4], and 17.4% [9]). monoterpene hydrocarbons such as β-pinene (38.2–40.5%) are dominant in *H. selaginifolium* oil [6,7]. Two chemical compositions are reported for *H. hypnoides* essential oil dominated by 1,8-cineole (51.5%) [11] or (E)-β-caryophyllene (34.0%) [6]. This last compound is also found as a major component in *H. cordifolium* oil (46.4–55.6%) [6,11] and in *H. russillonii* oil (29.5%) [11]. Finally, *H. bracteiferum* oil exhibited a chemical composition with β-pinene/1,8-cineole/α-humulene as major components [6,8,11]. These literature data reported an important chemical variability characterized by the presence of monoterpenes such as α-fenchene, β-pinene, 1,8-cineole or sesquiterpene hydrocarbons: (E)-β-caryophyllene, β-himachalene and α-humulene.

In this study, we were interested in six species: *Helichrysum dubardii* R. Vig. and Humbert, *H. benthamii* R. Vig. and Humbert, *H. hirtum* Humbert, *H. indutum* Humbert, *H. bojerianum* DC., *H. diotoides* DC [11]. *Helichrysum dubardii* and *H. benthamii* consist of subshrub plants with ericoid growth form, the leaves are deltoid, erect and applied on the twigs. *H. dubardii* leaves have only a single midrib vein, glabrous on the upper side, silvery white on the lower side. The bractal appendages of the inflorescences are yellowish white. In *H. benthamii*, the leaves have one to three veins arising from the base, covered with a dense gray tomentum above, loose underneath. The bractal appendages of the flowers are sulfur yellow [12]. *H. hirtum* and *H. indutum* are subshrub plants, leaves evenly distributed on the stem while the flowers have white bractal appendages. The first species has twigs covered with glandular hairs interspersed with fine cottony hairs and glands; the leaves are sessile, oblong. and with five veins arising from the base. The second species is covered with homogeneous fine cottony hairs sometimes dotted with sessile glands [12]. *H. bojerianum* and *H. diotoides* are also subshrub plants, with leaves evenly distributed on the stem. The bractal appendages of the inflorescences are sulfur yellow. *H. bojerianum* is covered with an ashy white aranose tomentum. The leaves are elliptical, acute sessile, with three veins arising from the base while *H. diotoides* is covered with a grayish aranose tomentum. The leaves are deltoid and sessile.

The aim of this work was to study for the first time, the chemical composition of the leaf essential oil extracted from these six endemic species growing wild in the center of Madagascar: *H. dubardii*, *H. benthamii, H. hirtum*, *H. indutum*, *H. bojerianum,* and *H. diotoides.*

## 2. Results

Twelve oil samples obtained by hydrodistillation (yields: 0.11–0.26%) of aerial parts of six *Helichrysum* species growing wild in Madagascar were analyzed by gas chromatography (GC) in combination with retention indices on two columns of different polarity, by gas chromatography coupled with mass spectroscopy (GC–MS) and by carbon-13 nuclear magnetic resonance (^13^C NMR). Due their complexity, *H. hirtum* and *H. dubardii* essential oils were also fractionated on silica gel column chromatography (CC). In total, 153 compounds were identified accounting for 90.8% to 99.9% of the total composition (Table 1).

### 2.1. Helichrysum Benthamii and H. dubardii Essential Oils

Two samples of *H. benthamii* (Hbe1 and Hbe2) produced a monoterpene hydrocarbon-rich oil characterized by the pre-eminence of α-pinene (50.8–51.9%), associated with sesquiterpene hydrocarbons: α-copaene (5.4–6.2%), α-humulene (3.1–4.4%) and (E)-β-caryophyllene (1.5–2.3%). The third sample of *H. benthamii* (Hbe3), also characterized by α-pinene (23.1%) as major compound, exhibited a slightly different chemical composition with percentages of sesquiterpene hydrocarbons more elevated: α-copaene (8.5%), α-humulene (6.4%) and (E)-β-caryophyllene (5.2%).

The main components of *H. dubardii* oil samples (Hd1-Hd3) were 1,8-cineole (26.9–35.7%), followed by α-pinene (5.8–6.6%), terpinen-4-ol (4.7–4.9%) and α-terpineol (4.0–4.3%). Sesquiterpene hydrocarbons were represented by α-muurolene (3.2–7.2%), γ-cadinene (1.6–3.5%), δ-cadinene (1.8–4.2%). It is noticeable that beyerene, a rare diterpene hydrocarbon was found in the three samples (0.2–0.5%).

### 2.2. Helichrysum Indutum, H. bojerianum and H. diotoides Essential Oils

*Helichrysum indutum*, *H. bojerianum* and *H. diotoides* (Hi, Hbo and Hdi samples) produced sesquiterpene-rich oils (47.9–70.2%): (E)-β-caryophyllene, ar- and γ-curcumenes, γ- and δ-cadinenes, aristolochene. Furthermore, among the monoterpenes, linalool was found in an appreciable amount in all samples (5.3–9.8%) whereas 1,8-cineole was identified only in *H. indutum* EO (13.4%). It could be pointed out that *H. diotoides* oil is characterized by the presence of several lilac derivatives: lilac alcohol A (2S,2′S,5′S) (1.7%), lilac alcohol B (2R,2′S,5′S) (3.6%) [13], lilac acetate A (0.6%), lilac acetate B (2.3%), lilac acetate C (0.3%), and lilac acetate D (8.7%).

### 2.3. Helichrysum Hirtum Essential Oil

The chromatographic profile of *H. hirtum* oil samples (Hh1, Hh2 and Hh3) varied drastically from the others and was characterized by the presence of many oxygenated sesquiterpenes. In the process of analyzing the chemical composition of the essential oils, we noticed that several compounds remained undetermined, providing very unsatisfactory matching with commercial or in- house MS libraries. Then, the EO was fractionated by silica gel column chromatography (CC), using a gradient of solvents. These compounds could however be identified from the fraction of CC by applying our in-house ^13^C NMR computerized methodology [14,15].

We highlighted the identification of presilperfolane and silphiperfolane derivatives as major components in the three samples by comparison of their carbon chemical shifts values with those reported in the literature [16,17,18,19,20,21,22]: 7-*epi*-silphiperfol-5-en-13-oic acid (4.0–20.8%) and silphiperfol-5-en-13-oic acid (4.0–10.6%) (Table 2), 7-*epi*-subergorgiol (7.6–14.8%), 7β-H-silphiperfol-5-ene (1.8–14.8%), presilphiperfolan-9-α-ol (6.9–8.0%), 7-*epi*-silphiperfolenal (1.4–2.5%), 13- hydroxysilphiperfol-6-ene (1.2–1.9%) and 7α-H-silphiperfol-5-ene (up to 0.3%). The presence of a compound including an acid group as major component is very unusual in essential oils.

We detailed the identification by ^13^C NMR of 7-*epi*-silphiperfol-5-en-13-oic acid (152) and silphiperfol-5-en-13-oic acid (153) in the sample Hh1 (Table 2). For these two compounds:-all the expected signals were observed;-the chemical shift variations between the reference spectrum (Marco et al. [16]) and the recorded spectrum of the sample Hh1 were low. Indeed, they were less than or equal to 0.08 ppm for at least 12 signals out of 15. Only the carbons of the acid function or near the acid function (i.e., C5, C6, C13) exhibited a higher chemical shift variation;-a DEPT sequence confirmed the number of hydrogens linked to each carbon.

It should be point out that the chemical shift values of carbons, measured on spectra recorded using high field spectrometers, were given with two decimal places. Nevertheless, it occasionally arose that chemical shift values were given with only one decimal. In such a case, although it is not mathematically correct, comparison of data given with one decimal and those given with two decimals unambiguously allowed identification of compounds.

## 3. Discussion

In a recent review, Rafidison et al., highlighed the actual state of Malagasy medicinal plants and particularly the pharmacological and ethnobotanical investigations. Croton and *Helichrysum* are the most cited genera. Even more, *H. faradafini* is present in the top 20 most cited species. Concerning essential oils, *H. faradafini*, *H. bracteiferum*, and *H. gymnocephalum* are actually the most produced in Madagascar and used as expectorant and as a preventative or curative remedy for treating coughs, colds, and bronchitis [2].

The studied oils reported several major components previously described in Malagasy *Helichrysum* EOs: (i) *H. dubardii* oil exhibited a close composition reported from *H. bracteiforum* oil and characterized by 1,8-cineole as major component (26.9–35.7% vs. 27.3% respectively) [6,11]; (ii) *H. indutum* EO composition is dominated by (E)-β-caryophyllene which is also reported to be in similar amounts in *H. faradifani* [6], *H. hypnoides* [6] and *H. russillonii* [11] EOs (33.1% vs. 34.6%, 34.0% and 29.5% respectively). Thus, *H. dubardii* and *H. indutum* EOs, which exhibited a chemical profile close to *H. bracteiferum* and *H. faradifani* respectively, were the good candidates for domestication.

*H. bojerianum* and *H. diotoides* EOs exhibited a different chemical composition. Even if, the percentage of (E)-β-caryophyllene was low in *H. bojerianum* and *H. diotoides* EOs (16.1% and 15.0% respectively), both oils can be classified as sesquiterpene hydrocarbon-rich oil (76.8% and 58.6% respectively). However, the *H. bojerianum* EO appeared original by the presence of six lilac derivatives (two alcohols and four acetates) at an appreciable ratio around 15%.

The composition of *H. benthamii* EO exhibited α-pinene as major component (23.1–51.9%) while β-pinene was frequently reported as the major component [6,7,8,11]. However, percentages up to 20% have never been observed in the *Helichrysum* genus.

Finally, *H. hirtum* EO can be classified as sesquiterpene hydrocarbon-rich oil (91.0%) but the chemical composition differed drastically from the others by the (i) absence of monoterpene hydrocarbons (only traces of α-pinene and 0.1% of p-cymenene), (ii) a very low amount of oxygenated monoterpenes (1.3%), (iii) the presence of several sesquiterpenes exbibiting silphiperfolane and presilphiperfolane skeletons. To our knowledge, the presence of silphiperfolane and presilphiperfolane derivatives has never been reported in Helychrysum EOs but these skeletons were reported in asteraceae family (*Petasites, Matricaria, Sphaeranthus, Otanthus*) [23].

This original chemical composition can be an important feature for marketing. Taking into account that the wild populations of *H. hirtum* were distributed in a limited area (Tapia or *Uapaca bojeri* Forest, highlands of Madagascar—around Arivonimamo), over-exploitation has greatly increased the vulnerability of *H. hirtum* [24]. Therefore, the protection of *H. hirtum* populations should be a high priority now and domestication can be considered as an excellent alternative to supply the continuous market needs by producing high quality and stable raw material, and at the same time, alleviating the pressure on natural resources from overharvesting.

## 4. Materials and Methods

### 4.1. Plant Material

Aerial parts of five *Helichrysum* species were collected in September 2016 (dry season) at the region of Itasy, district of Arivonimamo (Figure 1): *H. dubardii* (Ambatobe, 19°14′42.05″ S, 47°00′25.09″ E)*, H. benthamii* (Ambatobe, 19°14′42.05″ S, 047°00′25.09″ E)*, H. bojerianum* (Near Mount of Tsiafakafokely, 2203 m above sea level, 19°16′42.7″ S, 047°12′52.7″ E)*, H. diotoides* (South of Alakamisiy kely, 19°15′10.6″ S, 47°06′39.3″ E), *H. hirtum* (West of Arivonimamo, 19°01′85.09″ S, 47°16′90.9″ E). Aerial parts of *H. indutum* were collected in September 2019 (dry season) at region Alaotra Mangoro, District of Maromizaha (19°16′48.9′’ S, 047°02′05.2′’ E) (Figure 1). Voucher specimens were deposited at TAN and CNARP herbaria under the accession *Rakotonandrasana 1501* for *H. dubardii, 1502* for *H. benthamii, ST 1518* for *H. bojerianum, ST 1508* for *H. diotoides, ST 1519* for *H. hirtum* and, *ST 1520* for *H. indutum*.

The essential oils were obtained by hydrodistillation of fresh aerial parts (around 500–1000 g) over 3 h. Yields were calculated from fresh material: *H. dubardii,* 0.13–0.26%; *H. benthamii,* 0.12–0.21%; *H. bojerianum,* 0.26%; *H. diotoides,* 0.21%; *H. hirtum,* 0.11–0.18% and *H. indutum* 0.19%.

### 4.2. Gas Chromatography (GC) Analysis

GC analyses were performed on a Clarus 500 FID gas chromatograph (PerkinElmer, Courtaboeuf, France) equipped with two fused silica gel capillary columns (50 m, 0.22 mm, film thickness 0.25 m), BP-1 (polydimethylsiloxane) and BP-20 (polyethylene glycol). The oven temperature was programmed from 60 to 220 °C at 2 °C/min and then held isothermal at 220 °C for 20 min, injector temperature: 250 °C; detector temperature: 250 °C; carrier gas: hydrogen (1.0 mL/min); split: 1/60. The relative proportions of the oil constituents were expressed as percentages obtained by peak area normalization, without using correcting factors. Retention indices (RIs) were determined relative to the retention times of a series of n-alkanes with linear interpolation (‘Target Compounds’ software of PerkinElmer).

### 4.3. Mass Spectrometry

The EOs were analyzed with a PerkinElmer TurboMass detector (quadrupole, PerkinElmer, Courtaboeuf, France), directly coupled to a PerkinElmer Autosystem XL (PerkinElmer), equipped with a fused silica gel capillary column (50 m × 0.22 mm i.d., film thickness 0.25 µm), BP-1 (polydimethylsiloxane). Carrier gas, helium at 0.8 mL/min; split: 1/75; injection volume: 0.5 µL; injector temperature: 250 °C; oven temperature programmed from 60 to 220 °C at 2 °C/min and then held isothermal (20 min); ion source temperature: 250 °C; energy ionization: 70 eV; electron ionization mass spectra were acquired over the mass range 40–400 Da.

### 4.4. NMR Analysis

^13^C NMR analyses were performed on an AVANCE 400 Fourier Transform spectrometer (Bruker, Wissembourg, France) operating at 100.623 MHz for ^13^C, equipped with a 5 mm probe, in CDCl_3_, with all shifts referred to internal tetramethylsilane (TMS). ^13^C NMR spectra were recorded with the following parameters: pulse width (PW): 4 µs (flip angle 45°); acquisition time: 2.73 s for 128 K data table with a spectral width (SW) of 220.000 Hz (220 ppm); CPD mode decoupling; digital resolution 0.183 Hz/pt. The number of accumulated scans ranged from 2000–3000 for each sample (around 40 mg of oil in 0.5 mL of CDCl_3_). Exponential line broadening multiplication (1.0 Hz) of the free induction decay was applied before Fourier Transformation.

### 4.5. Identification of Individual Components

Identification of the components was based: (i) on comparison of their GC retention indices (RIs) on polar and apolar columns, determined relative to the retention times of a series of *n*-alkanes with linear interpolation (“Target Compounds” software of PerkinElmer), with those of authentic compounds and (ii) on comparison of the signals in the ^13^C NMR spectra of EOs with those of reference spectra compiled in the laboratory spectral library, with the help of a laboratory-made software [13,14,15]. In the investigated samples, individual components were identified by NMR at contents as low as 0.5%. Several compounds were identified by comparison of ^13^C NMR chemical shifts with those reported in the literature, for instance 7-epi silphiperfol-5-en-13-oic acid and silphiperfol-5-en-13-oic acid [16], beyerene [17], δ-terpineol [18], 7-epi-silphiperfolenal and 7- episubergorgiol [19], 13-hydroxysilphiperfol-6-ene [20], pogostol [21], 14-hydroxy-α-humulene [22], and lilac alcohol B [13].

### 4.6. Essential Oil Fractionation

*H. dubardii* oil sample Hd1 (1.0 g) was submitted to flash chromatography (silica gel: 200–500 µm). Four fractions (FHd1-FHd4) were eluted with a mixture of solvents of increasing polarity with pentane:diethyl ether (P:E) 100:0 to 0:100: FHd1 (P:E 100:0; 231.4 mg), FHd2 (P:E 98:2; 289.1 mg), FHd3 (P:E 95:5; 163.2 mg), and FHd4 (P:E 0:100; 218.4 mg). All fractions of chromatography were analyzed by GC (RI), GC–MS and ^13^C NMR.

An *H. hirtum* oil sample Hh1 (1.0 g) was also submitted to flash chromatography (silica gel: 200–500 µm). Six fractions (FHh1-FHh6) were eluted with a mixture of solvents of increasing polarity P:E 100:0 to 0:100: FHh1 (P:E 100:0; 143.1 mg), FHh2 (P:E 98:2; 23.0 mg); FHh3 (P:E 95:5; 156.5 mg), FHh4 (P:E 90:10; 524.2 mg), FHh5 (P:E 80:20, 126.0 mg) and FHh6 (P:E 0:100, 18.0 mg). All fractions of chromatography were analyzed by GC (RI), GC–MS, and ^13^C NMR.

## 5. Conclusions

This study provides useful scientific data to promote in situ conservation and to select chemical profiles for eventual production. Our results confirmed that Malagasy *Helichrysum* EOs exhibited an important chemical variability and these data are useful for projects of biodiversity conservation.

## Figures and Tables

**Figure 1 plants-09-00265-f001:**
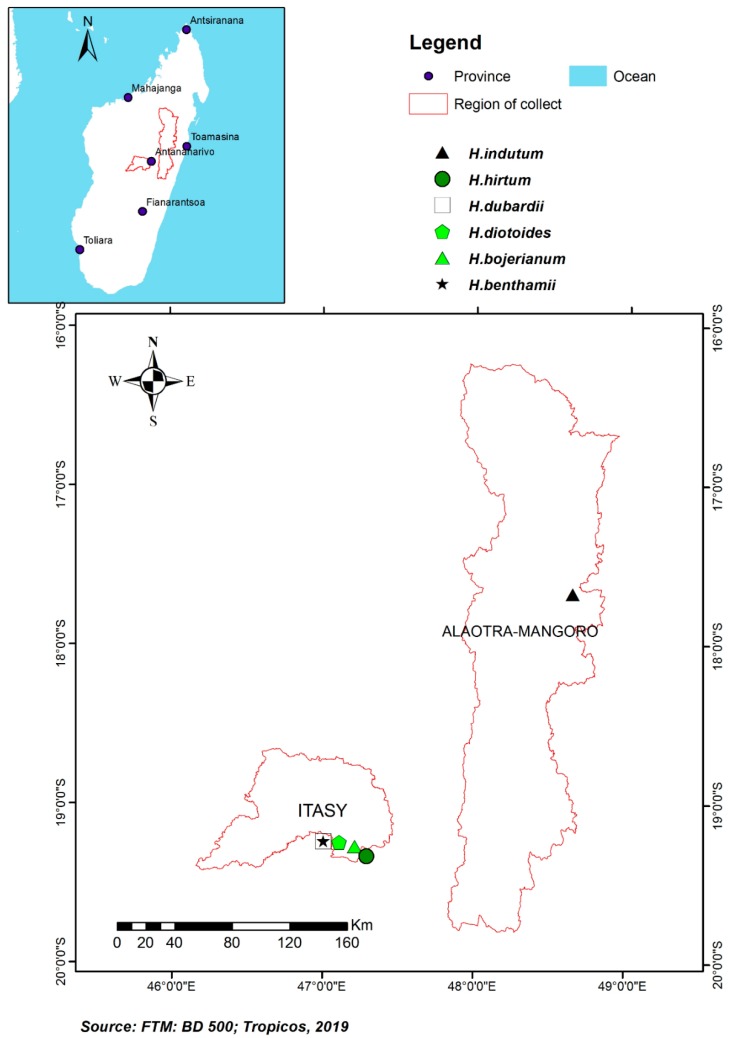
Collection maps of the six *Helichrysum* species.

**Table 1 plants-09-00265-t001:** Chemical composition of six *Helichrysum* essential oil samples.

No	Samples			*H. benthami*			*H. dubardii*			*H. ind*	*H. boj*	*H dio*	*H. hirtum*		
	Components	RI_a_	RI_p_	Hbe1	Hbe2	Hbe3	Hd1	Hd2	Hd3	Hi	Hbo	Hdi	Hh1	Hh2	Hh3
1	α-Thujene	920	1014	0.2	0.2	-	0.3	0.4	0.3	-	-	-	-	-	-
2	α-Pinene	929	1014	**50.8**	**51.9**	**23.1**	**5.8**	**6.6**	**5.8**	**2.7**	tr	tr	**1.6**	-	tr
3	α-Fenchene	941	1053	0.1	0.1	tr	-	-	tr	-	-	-	-	-	-
4	Camphene	943	1063	0.3	0.2	0.1	0.3	0.3	0.3	0.3	-	-	-	-	-
5	Thuja-2,4(10)-diene	944	1125	0.7	0.4	0.2	-	-	0.1	-	-	-	-	-	-
6	Oct-1-en-3-ol	961	1445	-	-	-	-	-	-	0.2	-	-	-	-	-
7	Sabinene	964	1121	0.6	**1.0**	**0.7**	**0.7**	**0.6**	**0.7**	-	-	-	0.1	-	-
8	β-Pinene	969	1110	0.6	**0.7**	**0.6**	**2.8**	**2.1**	**2.8**	**0.8**	-	-	0.1	-	-
9	1,8-dehydro Cineole	976	1192	0.1	0.1	tr	0.2	0.1	0.2	-	-	-	-	-	-
10	Octanal	976	1281	0.1	tr	tr	-	-	-	-	-	-	-	-	-
11	Myrcene	979	1159	0.1	0.1	tr	**0.6**	0.4	0.6	**1.3**	0.1	-	0.2	-	-
12	α-Phellandrene	995	1164	0.2	0.1	0.1	0.1	0.2	0.2	tr	-	-	-	-	-
13	α-Terpinene	1008	1179	0.2	0.2	0.2	**1.2**	**1.3**	**1.2**	0.2	tr	-	-	-	-
14	*p*-Cymene	1010	1269	**0.6**	**0.6**	0.4	**1.5**	**1.0**	**1.5**	**0.8**	0.1	tr	-	-	-
15	Limonene	1019	1200	**1.6**	0.4	**1.1**	**0.7**	0.4	**0.7**	**5.4**	**1.2**	0.4	0.1	-	-
16	1,8-Cineole	1021	1212	**3.0**	**4.0**	**3.0**	**28.2**	**35.7**	**26.9**	**13.4**	**1.7**	**0.5**	0.1	-	tr
17	(*Z*)-β-Ocimene	1023	1232	tr	tr	0.1	-	-	-	**0.5**	0.1	-	-	-	-
18	(*E*)-β-Ocimene	1036	1249	tr	**-**	tr	-	-	tr	**1.9**	0.2	-	-	-	-
19	γ-Terpinene	1047	1243	0.3	0.3	0.3	**2.6**	**2.5**	**2.6**	**1.0**	0.3	0.1	-	-	-
20	*trans*-Sabinene hydrate	1050	1462	0.1	-	tr	0.1	-	0.2	-		-	-	-	-
21	*cis*-Linalool oxide THF	1055	1442	0.1	-	-	-	-	-	-	tr	-	-	-	-
22	Nonan-2-one	1067	1388	0.3	0.2	-	-	-	-	-	0.1	0.1	-	-	-
23	*p*-Cymenene	1070	1438	0.2	0.1	-	0.1	0.1	0.1	-	0.1	-	-	-	0.1
24	Terpinolene	1076	1283	0.2	0.1	-	**0.6**	**0.6**	**0.6**	0.3	0.3	-	-	-	**-**
25	Nonanal	1079	1394	0.2	0.2	-	0.1	**-**	0.1	-	-	-	-	-	**-**
26	Linalool	1082	1544	0.2	0.1	-	**0.9**	0.4	0.1	**7.2**	**5.3**	**9.8**	-	-	0.4
27	Fenchyl alcohol	1099	1584	-	-	-	-	-	-	0.3	0.4	0.3	-	-	-
28	α-Campholenal	1101	1482	**1.0**	**0.6**	**1.0**	-	-	tr	-	-	-	-	-	-
29	*cis-p*-Menth-2-en-1-ol	1104	1482	0.1	0.1	0.1	0.2	0.2	0.2	-	-	-	-	-	-
30	Camphor	1118	1515	-	-	-	tr	0.1	tr	-	-	-	-	-	-
31	*trans*-Pinocarveol	1122	1653	**1.4**	**1.0**	**1.8**	0.3	**0.6**	**-**	0.2	-	-	-	-	-
32	*trans*-Verbenol	1126	1677	**0.6**	0.3	0.2	0.1	-	0.1	-	0.1	-	-	-	0.1
33	Camphene hydrate	1133	1592	-	tr	tr	-	-	tr	0.1	-	-	-	-	-
34	Pinocarvone	1135	1567	**0.7**	**0.6**	**1.0**	-	0.2	-	-	0.1	0.1	-	-	-
35	*p*-Mentha-1,5-dien-8-ol	1143	1714	**1.2**	-	-	-	-	-	-	-	-	-	-	-
36	δ-Terpineol	1144	1670	-	**0.7**	**0.9**	0.4	0.5	**0.4**	0.2	0.1	tr	-	-	-
37	Borneol	1146	1682	**2.4**	**0.8**	**2.1**	**1.7**	**1.5**	**1.7**	**0.7**	1.4	**0.8**	-	-	-
38	Mentha-1,8-dien-4-ol	1156	1673	-	-	-	-	-	0.1	0.1	0.1	0.1	-	-	-
39	*p*-Cymen-8-ol	1157	1848	0.2	0.2	0.2	-	-	-	-	-	-	-	-	-
40	Terpinen-4-ol	1160	1599	0.4	**0.7**	**0.9**	**4.9**	**4.7**	**4.9**	**1.0**	**0.7**	0.3	-	-	**0.6**
41	Myrtenal	1167	1627	**0.5**	0.3	**0.5**	-	-	0.1	0.0	-	-	-	-	-
42	α-Terpineol	1170	1694	0.4	**0.5**	**0.5**	**4.3**	**4.0**	**4.3**	**1.8**	**2.5**	**1.5**	-	-	-
43	Myrtenol *	1177	1790	-	0.4	0.3	-	tr	0.2	0.2	-	-	-	-	-
44	α-Campholenol *	1177	1791	**0.9**	-	0.4	0.2	**-**	-	-	-	-	-	-	-
45	Lilac alcohol A (2S,2′S,5′S)	1179	1742	-	-	-	-	-	-	-	-	**1.7**	-	-	-
46	Verbenone	1182	1703	0.1	0.2	0.3	-	-	tr	-	-	-	-	-	-
47	*cis*-Piperitol	1184	1682	tr	tr	-		-	tr	-	-	-	-	-	-
48	Lilac alcohol B (2R,2′S,5′S)	1188	1718	-	-	-	-	-	-	-	-	**3.6**	-	-	-
49	*trans*-Carveol	1196	1834	0.3	0.1	0.2	-	-	tr	0.2	tr	-	-	-	-
50	Nerol	1208	1799	-	-	-	-	-	tr	-	tr	tr	-	-	-
51	Carvone	1212	1738	0.1	0.1	0.1	-	-	-	tr	**1.3**	-	-	-	-
52	Piperitone	1233	1730	-	-	-	-	-	-	-	tr	-	-	-	-
53	Geraniol	1234	1835	-	-	-	-	-	-	0.1	-	0.1	-	-	-
54	Isopiperitenone	1236	1859	-	-	-	-	-	-	-	0.1	-	-	-	-
55	Bornyl acetate	1266	1577	-	-	tr	0.5	0.3	0.5	tr	**-**	0.2	-	-	-
56	Lavandulyl acetate	1269	1605	-	-	-	-	-	-	0.1	0.2	**0.6**	-	-	-
57	Myrtenyl acetate	1305	1678	0.1	tr	0.1	0.1	-	0.1	0.2	-	-	-	-	0.2
58	Piperitenone	1306	1910	-	-	-	-	-	-	-	-	0.2	-	-	-
59	Lilac acetate A ^#^	1318	1773	-	-	-	-	-	-	-	0.2	**0.6**	-	-	-
60	Lilac acetate B ^#^	1322	1744	-	-	-	-	-	-	-	-	**2.3**	-	-	-
61	7α-*H*-Silphiperfol-5-ene	1324	1420	-	-	-	-	-	-	-	-	-	0.1	-	0.3
62	Eugenol	1325	2165	0.1	0.1	0.2	-	-	tr	-	-	-	-	-	-
63	Lilac acetate C ^#^	1328	1779	-	-	-	-	-	-	-	-	0.3	-	-	-
64	Lilac acetate D ^#^	1333	1773	-	-	-	-	-	-	-	-	**8.7**	-	-	-
65	7β-*H*-Silphiperfol-5-ene	1343	1446	-	-	-	-	-	-	-	-	-	**9.1**	**1.8**	**14.8**
66	α-Cubebene	1346	1453	0.1	0.2	0.3	**1.2**	**0.6**	**1.2**	-	-	-	0.3	-	0.4
67	Clovene	1363	1501	**0.5**	0.4	-	-	-	0.2	-	0.2	0.1	-	-	-
68	Cyclosativene	1367	1476	0.2	0.1	0.2	-	-	tr	-	**-**	-	**0.9**	0.1	**1.4**
69	α-Ylangene	1371	1484	-	-	-	-	-	-	-	0.1	-	-	-	-
70	α-Copaene	1372	1486	**5.4**	**6.2**	**8.5**	**2.3**	**1.8**	**2.3**	**0.8**	**2.0**	**1.1**	**0.5**	0.1	**1.4**
71	β-Bourbonene	1380	1514	0.4	0.4	**0.8**	0.2	0.2	0.2	-	-	-	**-**	-	-
72	β-Cubebene	1384	1532	0.2	0.2	0.4	0.3	-	-	-	-	-	**-**	-	0.1
73	Sativene	1388	1528	0.1	0.2	0.2	-	-	-	-	tr	-	0.3	-	0.3
74	Ylanga-2,4(15)-diene	1397	1606	0.5	0.6	0.7	-	-	-	-	-	-	-	-	-
75	Italicene	1400	1537	-	-	-	-	-	-	-	**1.6**	**1.5**	-	-	-
76	Isocaryophyllene	1407	1570	-	0.1	0.2	-	-	-	0.2	0.2	0.1	-	-	-
77	*cis*-α-Bergamotene	1409	1562	tr	tr	0.4	-	-	-	-	0.2	0.3	-	-	-
78	(*E*)-β-Caryophyllene	1415	1590	**1.5**	**2.3**	**5.2**	**2.7**	**3.4**	**2.7**	**33.1**	**16.1**	**15.0**	**3.5**	**0.8**	**3.7**
79	β-Copaene	1422	1585	0.1	0.1	0.2	0.2	**-**	0.2	**-**	tr	-	**-**	-	0.1
80	*trans*-α-Bergamotene	1429	1578	0.1	0.1	0.2	-	-	-	tr	0.2	0.1	**-**	-	tr
81	α-Guaiene	1433	1583	-	-	0.1	-	-	tr	0.3	0.3	-	0.2	-	0.1
82	*trans*-Muurola-3,5-diene	1441	1746	0.1	0.1	0.2	-	-	0.2	0.1	-	-	-	-	-
83	Aromadendrene	1443	1585	-	-	-	-	-	-	-	0.2	0.2	-	-	-
84	α-Humulene	1448	1662	**3.1**	**4.4**	**6.4**	**1.6**	**0.5**	**1.6**	**3.3**	**2.3**	**3.5**	**4.7**	**1.1**	**5.1**
85	(*E*)-9-*epi*-Caryophyllene	1452	1637	0.2	0.1	0.1	-	-	-	-	0.5	0.3	-	-	-
86	α-Acoradiene	1457	1669	tr	tr	0.1	-	-	-	-	0.1	0.1	-	-	-
87	β-Acoradiene	1460	1689	0.1	tr	tr	-	-	0.1	-	0.2	0.1	-	-	-
88	*trans*-Cadina-1(6),4-diene	1466	1654	0.1	0.1	0.2	-	-	1.5	0.6	0.2	-	-	-	0.6
89	*ar*-Curcumene	1468	1756	-	-	-	-	-	-	**2.9**	**2.8**	-	-	-	**2.9**
90	γ-Muurolene	1469	1686	0.1	**1.4**	**1.9**	**1.5**	**1.0**	0.3	**0.5**	**-**	**3.8**	0.6	**-**	**1.1**
91	γ-Curcumene	1473	1686	0.1	0.1	0.2	**-**	**-**	-	-	**9.0**	-	**-**	**-**	**-**
92	*trans*-β-Bergamotene	1475	1676	0.1	0.1	0.1	**-**	**-**	-	-	-	-	**-**	**-**	**-**
93	Selina-4,11-diene	1476	1669	-	-	-	**-**	**-**	-	-	0.3	**1.0**	**-**	**-**	**-**
94	β-Selinene	1477	1708	0.4	0.2	0.4	**1.9**	**2.4**	**1.9**	0.5	**2.5**	**2.6**	**-**	**-**	**-**
95	Aristolochene	1478	1697	-	-	-	**-**	**-**	-	-	**6.2**	-	**-**	**-**	**-**
96	Germacrene D	1481	1707	0.2	0.2	0.4	-	-	0.2	-	-	-	-	-	0.1
97	α-Selinene	1487	1718	0.2	0.1	0.4	**1.3**	**0.8**	**1.3**	-	-	-	-	-	**-**
98	*epi*-Zonarene	1488	1706	-	-	-	-	-	-	-	4.5	3.4	-	-	**-**
99	α-Muurolene	1489	1718	**0.6**	**0.9**	**1.2**	**3.2**	**7.2**	**3.2**	**1.0**	**2.3**	**1.1**	0.2	-	**0.7**
100	Amorpha-4,7(11)-diene	1490	1723	-	-	-	-	-	-	**1.1**	**-**	-	-	-	**-**
101	(*E,E*)-α-Farnesene	1494	1743	-	-	-	-	-	-	-	-	-	0.2	-	0.3
102	β-Bisabolene *	1497	1734	0.1	0.1	tr	-	-		-	-	**0.6**	-	-	-
103	α-Bulnesene *	1497	1709	-	-	-	-	-	**0.8**	0.2	0.3	-	0.3	-	**0.6**
104	β-Curcumene	1500	1734	-	-	-	-	-	-	-	**0.9**	0.4	-	-	-
105	γ-Cadinene	1502	1752	0.1	0.2	**0.5**	**3.5**	**1.6**	**3.5**	**1.7**	**6.5**	**4.6**	-	-	**0.6**
106	*cis*-Calamene	1506	1826	0.2	0.3	0.3	-	-	-	0.2	0.4	0.4	-	-	**0.5**
107	Presilphiperfolan-9-α-ol	1510	2006	-	-	-	-	-	-	-	-	-	**8.0**	***7.0***	***6.9***
108	δ-Cadinene	1512	1750	**1.5**	**1.7**	**3.0**	**4.2**	**1.8**	**4.2**	**2.4**	**7.6**	**5.6**	**1.3**	0.3	**0.9**
109	Zonarene	1515	1710	-	-	-	-	-	-	0.3	**1.0**	**0.6**	-	-	0.2
110	*trans*-Cadina-1,4-diene	1521	1712	0.1	0.1	0.1	-	-	-	0.2	**0.5**	0.4	-	-	-
111	α-Calacorene	1524	1909	0.4	0.3	**0.7**	0.2	-	0.2	-	-	0.2	-	-	0.2
112	α-Cadinene	1525	1783	-	-	-	-	-	0.2	0.1	**0.9**	**0.7**	-	-	-
113	Selina-4(15),7(11)-diene	1529	1737	-	-	-	-	-	-	0.3	-	-	-	-	-
114	(*E*)-α-Bisabolene	1530	1762	-	-	-	-	-	-	-	0.1	0.1	-	-	-
115	Selina-3,7(11)-diene	1538	1737	-	-	-	-	-	-	0.1	-	-	-	-	-
116	(E)-Nerolidol	1545	2039	tr	tr	tr	-	-	-	-	tr	-	0.2	-	0.3
117	β-Calacorene	1547	1883	0.1	0.1	0.6	-	-	-	0.1	-	-	-	-	-
118	Caryolan-8-ol	1556	2046	tr	-	-	-	-	-	**1.4**	**1.2**	**2.1**	-	-	0.2
119	Spathulenol	1559	2118	0.4	0.1	**1.2**	0.1	-	0.1	-	-	-	-	-	-
120	Caryophyllene oxide	1567	1974	**1.5**	**3.6**	**6.8**	**1.1**	**1.3**	**1.1**	**2.4**	0.2	0.2	**3.7**	**2.2**	**4.8**
121	Gleenol	1568	2029	0.3	0.1	0.2	**2.0**	**2.0**	**2.0**	-	-	tr	-	-	-
122	7-epi-Silphiperfolenal	1573	2024	-	-	-	-	-	-	-	-	-	***2.5***	***1.4***	***2.3***
123	Humulene oxide I	1580	2009	0.2	-	0.3	-	-	-	-	-	0.1	-	-	0.4
124	β-Oplopenone	1585	2064	-	-	-	0.1	-	0.1	-	-	-	-	-	-
125	*epi*-Cubenol	1590	2056	-	-	-	**0.5**	0.4	**0.5**	-	-	**0.8**	-	-	-
126	Globulol *	1591	2058	-	-	0.6	-	-	-	-	-	-	-	-	-
127	Humulène oxide II *	1591	2031	**1.2**	**3.6**	**3.6**	-	-	0.3	0.4	0.1	0.2	**2.4**	**1.5**	**3.4**
128	Copaborneol*	1591	2169	**1.2**	***0.9***	**1.2**	0.1	0.3	**0.1**	-	-	0.2	***1.0***	***1.0***	***1.0***
129	7-epi-Subergorgiol	1598	2252	-	-	-	-	-	-	-	-	-	***14.8***	***13.1***	***7.6***
130	Muurola-4,10(14)-dien-1-β-ol	1606	2142	0.3	0.3	**0.8**	0.4	0.4	***1.5***	-	**-**	-	-	-	-
131	Cadina-4,10(14)-dien-1-α-ol	1608	2143	0.3	-	-	-	-	-	-	0.2	0.4	-	-	0.4
132	1,10-di-epi-Cubenol	1610	2058	0.2	0.4	**0.5**	-	-	**0.5**	-	**0.4**	***0.5***	-	-	0.2
133	Caryophylla-4(12),8(13)-dien-5β-ol	1616	2284	0.1	tr	**0.7**	0.3	0.4	0.3	-	-	-	-	-	0.1
134	τ-Cadinol *	1622	2164	-	-	-	0.4	0.2	0.4	**1.2**	**2.4**	**2.2**	-	-	0.1
135	τ-Muurolol *	1622	2180	-	0.1	0.2	**1.0**	0.3	**1.0**	0.2	0.4	**0.5**	-	-	0.1
136	β-Betulenal *	1626	2151	-	-	-	-	-	-	-	-	-	***5.4***	***3.9***	***3.8***
137	α-Muurolol *	1626	2190	0.1	0.2	0.2	-	-	0.1	0.1	0.1	0.1	-	-	-
138	Cubenol	1628	2070	-	0.1	0.1	-	-	**-**	-	-	0.1	-	-	-
139	β-Eudesmol	1630	2222	-	-	-	0.1	-	**-**	0.1	0.2	-	-	-	-
140	α-Cadinol	1633	2225	-	-	-	**0.4**	-	**0.9**	0.3	**-**	-	-	-	-
141	Pogostol *	1635	2199	-	-	-	-	-	-	-	-	-	***0.9***	***0.8***	***0.6***
142	13-Hydroxysilphiperfol-6-ene *	1635	2289	-	-	-	-	-	-	-	-	-	***1.5***	***1.9***	***1.2***
143	4-α-Hydroxy-agarofuran *	1635	2211	-	-	-	-	-	-	-	-	**0.9**	-	-	-
144	α-Eudesmol	1638	2212	-	-	-	-	0.2	-	0.2	**0.8**	-	-	-	-
145	Intermedeol	1639	2226	-	-	-	-	-	-	**-**	**0.5**	**2.0**	-	-	-
146	14-hydroxy-β-Caryophyllene	1648	2323	-	-	-	-	-	-	**-**	**-**	-	***3.3***	***3.1***	***1.7***
147	α-Bisabolol	1664	2211	-	-	-	-	-	-	0.1	0.1	0.4	**-**	**-**	**-**
148	14-hydroxy-α-Humulene	1691	2448	-	-	-	-	-	-	-	-	-	***2.9***	***3.1***	***1.4***
149	7,14-anhydro-Amorpha-4.9-diene	1744	2522	-	-	-	-	-	1.0	-	-	-	-	-	-
150	Beyerene	1922	2184	-	-	-	**0.5**	0.2	**0.5**	-	-	-	-	-	-
151	Manool	2034	2646	***0.6***	tr	***0.9***	**-**	**-**	***-***	-	-	-	-	-	-
152	7-epi-Silphiperfol-5-en-13-oic acid	NE	2830	-	-	-	-	-	-	-	-	-	***18.2***	***40.0***	***20.8***
153	Silphiperfol-5-en-13-oic acid	NE	2927	-	-	-	-	-	-	-	-	-	***4.0***	***10.6***	***4.8***
Monoterpene Hydrocarbon				56.7	56.4	26.9	17.3	16.5	17.5	15.2	2.4	0.5	2.1	-	0.1
Oxygenated Monoterpene				13.9	10.8	13.6	42.1	48.3	40.0	25.8	14.2	31.7	0.1	-	1.3
Sesquiterpene Hydrocarbon				16.9	21.4	34.2	24.3	21.3	26.0	50.2	70.2	47.9	22.2	4.2	36.4
Oxygenated Sesquiterpene				10.3	9.4	16.4	6.5	5.5	9.9	6.4	6.6	10.7	68.8	89.6	62.1
Diterpene				0.6	-	0.9	0.5	0.2	0.5	-	-	-	-	-	-
Phenyl propanoid				0.1	0.1	0.2	-	-	tr	-	-	-	0.1	-	-
Acyclic compound				0.6	0.4	-	0.1	-	0.1	0.2	0.1	0.1	-	-	-
TOTAL				99.1	98.5	92.2	90.8	91.8	94.0	97.8	93.5	90.9	93.3	93.8	99.9

Order of elution and relative percentages of individual components are given on an apolar column (BP-1) excepted those with an asterisk (*) percentages on polar column (BP-20); RI_a_, RI_p_: retention indices measured on apolar and polar capillary columns respectively; percentages in bold: components identified by a combination of GC(RI), GC–MS and ^13^C NMR; *^13^C NMR (italic):* compounds identified by ^13^C NMR in CC fractions; tr: trace level (<0.05%); ^#^ isomer not determined; NE: compound non-eluted on an apolar column BP-1; *H. ind: H. indutum*, *H. boj: H. bojerianum, H. dio: H. diotoides.*

**Table 2 plants-09-00265-t002:** ^13^C NMR data (400 MHz, CDCl_3_) of compounds 152 and 153.

7-*epi*-silphiperfol-5-en-13-oic Acid	C ^1^	δC ppm ^2^	δC ppm ^3^	Δδ ^4^
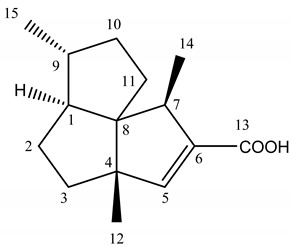	1	52.4	52.45	0.05
	2	30.4	30.41	0.01
	3	37.3	37.27	0.03
	4	57.6	57.66	0.06
	5	154.8	155.15	0.35
	6	136.9	136.74	0.16
	7	47.8	47.81	0.01
	8	65.3	65.37	0.07
	9	42.5	42.52	0.02
	10	35.9	35.96	0.06
	11	34.6	34.63	0.03
	12	19.7	19.69	0.01
	13	171.0	171.16	0.16
	14	14.8	14.83	0.03
	15	19.2	19.23	0.03
**silphiperfol-5-en-13-oic acid**	**C ^1^**	**δC ppm ^2^**	**δC ppm ^3^**	**Δδ ^4^**
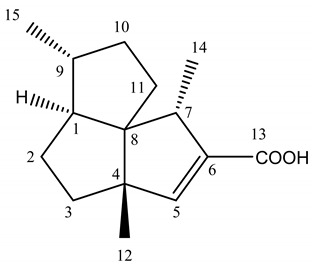	1	64.5	64.50	0.00
	2	30.0	30.05	0.05
	3	37.5	37.49	0.01
	4	58.6	58.68	0.08
	5	154.3	154.54	0.24
	6	138.6	138.63	0.03
	7	49.9	49.90	0.00
	8	64.0	64.04	0.04
	9	43.1	43.17	0.07
	10	35.6	35.64	0.04
	11	28.9	28.91	0.01
	12	19.9	19.90	0.00
	13	170.7	170.82	0.12
	14	18.0	18.02	0.02
	15	21.7	21.69	0.01

^1^ numbering according to Marco et al. [16]; ^2^ literature data; ^3^ experimental data; ^4^ differences between literature and experimental data.

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
