# Peer review of "Chemical Composition of Aerial Parts Essential Oils from Six Endemic Malagasy Helichrysum Species"

_plants, 2020, doi:10.3390/plants9020265_

Round 1

Reviewer 1 Report

Manuscript presents extensive and interesting work.In my opinion the Authors should add information on potential uses of Helichrysum species and the actions of their main ingredients (e.g. medical application??). In addition in my opinion the Discussion Section might be more extended.

Author Response

Dear Editor,

We took into account all the suggestions and requirements of reviewers. We revised the manuscript accordingly with changes highlighted in yellow.

Reviewer 1

Although the paper presents a few interesting data, the Discussion section is poorly written.

I think the authors should highlight the innovative elements along with the significance of their study, by discussing the potential beneficial effects of the main components of the essential oils and by correlating their concentration with the prevention and/or treatment of several disorders

Reviewer 2

Manuscript presents extensive and interesting work. In my opinion the Authors should add information on potential uses of Helichrysum species and the actions of their main ingredients (e.g. medical application??). In addition in my opinion the Discussion Section might be more extended.

Reviewer 3

The presented paper is interesting, but it is not acceptable for publication in its present form. Major shortcoming is lack of discussion. This sections covers only half pages and is based only on two papers. There are also no conclusions. Authors should post information on the importance of their research and its potential use.

The Discussion Section have been extended, as suggested by the three referees.

Three references [2], [23] and [24], have been added to enlarge the discussion.

A Conclusion Section has been added.

Additional reference [2] concerning the uses of Helichrysum species as medicinal plants including pharmacological and ethnobotanical investigations was introduced.

Rafidison V., Ratsimandresy, F., Rakotondrajaona R., Rasamison, V., Rakotoarisoa, M., Rakotondrafara, A., Rakotonandrasana, S.R. Synthèse et analyse de données sur les inventaires de plantes médicinales de Madagascar. Ethnobot. Res. Appl. 2019, 18, 1-19.

We hope that this revised manuscript, is well suited for publication in Plants.

Best regards,

Félix Tomi

Reviewer 2 Report

Although the paper presents a few interesting data, the Discussion section is poorly written.

I think the authors should highlight the innovative elements along with the significance of their study, by discussing the potential beneficial effects of the main components of the essential oils and by correlating their concentration with the prevention and/or treatment of several disorders.

In addition, possible applications of the essential oils studied in the food/pharmaceutical sector should be pointed out.

Author Response

(The authors gave the same response as above.)

Reviewer 3 Report

The presented paper is interesting, but it is not acceptable for publication in its present form. Major shortcoming is lack of discussion. This sections covers only half pages and is based only  on two papers. There are also no conclusions. Authors should post information on the importance of their research and its potential use.

Author Response

(The authors gave the same response as above.)

Round 2

Reviewer 3 Report

The authors corrected the manuscript as suggested. In its current form it can be published in Plants.